# Combined Merkel Cell Carcinoma with Nodal Presentation: Report of a Case Diagnosed with Excisional but Not Incisional Biopsy and Literature Review

**DOI:** 10.3390/diagnostics13030449

**Published:** 2023-01-26

**Authors:** Chih-Yi Liu, Nai-Wen Kang, Kengo Takeuchi, Shih-Sung Chuang

**Affiliations:** 1Division of Pathology, Sijhih Cathay General Hospital, New Taipei City 221, Taiwan; 2School of Medicine, College of Medicine, Fu Jen Catholic University, New Taipei City 221, Taiwan; 3Division of Hemato-Oncology, Department of Internal Medicine, Chi-Mei Medical Center, Tainan 710, Taiwan; 4Division of Pathology, Cancer Institute, Japanese Foundation for Cancer Research, Tokyo 135-8550, Japan; 5Pathology Project for Molecular Targets, Cancer Institute, Japanese Foundation for Cancer Research, Tokyo 135-8550, Japan; 6Department of Pathology, Cancer Institute Hospital, Japanese Foundation for Cancer Research, Tokyo 135-8550, Japan; 7Department of Pathology, Chi-Mei Medical Center, Tainan 710, Taiwan

**Keywords:** cytology, Merkel cell carcinoma, Merkel cell polyomavirus, paranuclear blue inclusion, skin cancer, thyroid transcription factor-1

## Abstract

Merkel cell carcinoma (MCC) is a rare primary neuroendocrine carcinoma (NEC) of the skin. As compared to pure MCCs, combined MCCs are aggressive and exhibit a higher probability of metastasis. A correct diagnosis might be missed, especially when the biopsy sample is too small or too superficial. We report a 79-year-old Taiwanese male who presented with lymphadenopathy suspicious for lymphoma. A nodal biopsy showed metastatic NEC. A skin tumor in the lower back was identified, and an incisional biopsy showed only squamous cell carcinoma (SCC). A subsequent excisional biopsy was performed based on the advice of the senior pathologist because of the presence of metastatic nodal NEC. Finally, a diagnosis of combined MCC and SCC was confirmed. Our literature review identified 13 cases of combined MCC with nodal metastasis as initial presentations, all with an aggressive clinical course. Both the MCC and non-MCC components could be present in the metastatic nodes. Metastases of pure MCC cells were observed in three combined MCCs in sun-protected areas, probably pointing to a distinct pathogenesis. Excision or punch biopsy to include the deep dermal NEC component is recommended as timely diagnosis is mandatory for appropriate management of patients with this rare skin cancer.

## 1. Background

Merkel cell carcinoma (MCC) is an aggressive primary cutaneous carcinoma with both epithelial and neuroendocrine features, mainly affecting the elderly [1,2,3]. The most frequent anatomic sites of MCC are the sun-exposed areas of the head and neck, followed by the extremities [3,4]. The well-documented risk factors include chronic sun exposure, UVA photochemotherapy, white skin type, male sex, immunosuppression, and a history of other skin cancers [2,3]. Surgery followed by adjuvant radiation therapy is the first-line treatment for localized MCC [3,5]. Chemotherapy has been used to treat advanced disease, while evidence on the efficacy of immunotherapy is still limited [3,5].

Divergent differentiation in MCC, although rare, has been reported in previous case series [6,7,8,9]. Combined MCCs account for 5–20% of all MCCs, most commonly in association with an invasive or in situ SCC [10]. Recently, the squamous and neuroendocrine components of combined MCCs were found to have overlapping genetic alterations, either originating from a keratinocytic precursor lesion or from tumor stem cells [10]. Compared to pure MCC, there is a higher frequency of metastasis and tumor-related death from combined MCC [6,8,9]. It is important to note that the diagnosis of combined MCCs could be missed if the MCC component is not detected in the incisional biopsy because of the limited sample size.

**Figure 1 diagnostics-13-00449-f001:**
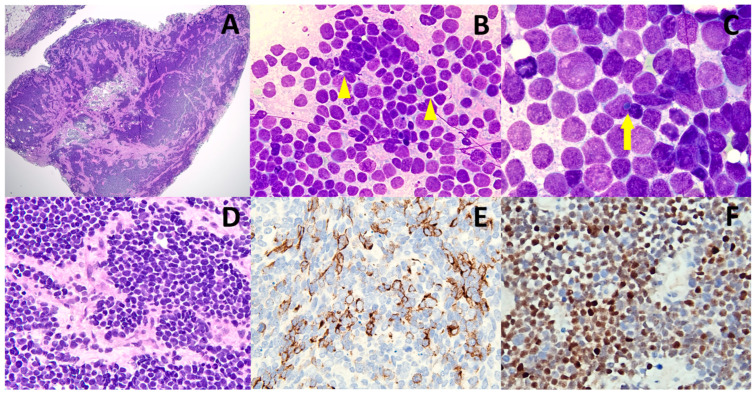
A 79-year-old Taiwanese male presented with a gradually enlarging left inguinal mass for two months. His past medical history was negative for immunosuppression or cutaneous malignancy. An abdominal CT scan revealed multiple abdominal lymphadenopathy suspicious for lymphoma. (**A**) An inguinal lymph node biopsy showed complete replacement by metastatic hyperchromatic tumor cells (magnification ×20). (**B**) Imprint cytology of the lymph node disclosed small-sized neoplastic cells exhibiting an “Indian” filing pattern (arrowhead) and nuclear molding, suggestive of small cell carcinoma or NEC (Liu stain, ×400) [11]. (**C**) A Romanowsky-stained slide depicted occasional paranuclear blue inclusions (arrow), supporting the diagnosis of NEC or MCC cells (Liu stain, ×1000) [11]. (**D**) A histologic slide showed cohesive tumor cells with round-shaped nuclei and a fine chromatin pattern (magnification ×200). (**E**,**F**) An immunohistochemical study demonstrated a perinuclear and/or dot-like staining pattern for CK20 (**E**) and a nuclear staining pattern for INSM1 (**F**). ((**E**,**F**), ×400). The inguinal lymph node biopsy was diagnosed as metastatic NEC.

**Figure 2 diagnostics-13-00449-f002:**
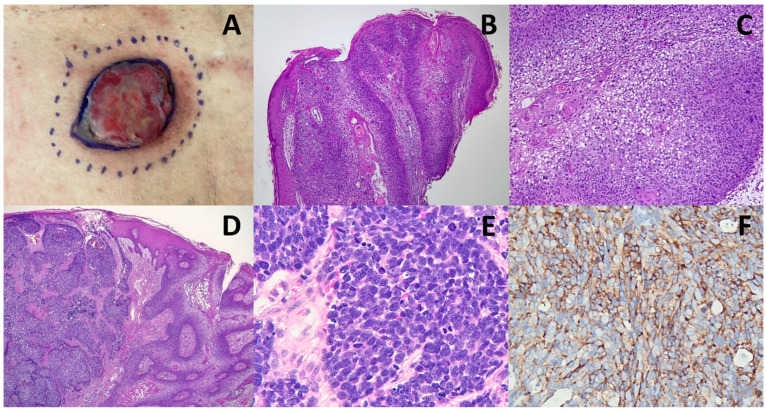
(**A**) Since there were no lung lesions by imaging studies, making small cell carcinoma unlikely, we examined his skin for MCC and found an erythematous, nodular lesion in the lower back, measuring 3 cm in diameter. (**B**,**C**) However, an incisional biopsy of the cutaneous tumor showed only well-differentiated SCC ((**B**), ×20), formed by polygonal squamous epithelial cells with keratinization ((**C**), ×100). (**D**–**F**) With the presentation of metastatic nodal NEC suggestive of MCC, one of us (S.-S.C.) urged the clinician for an excisional biopsy, which showed a well-differentiated SCC on the surface (right-hand side; (**D**)) and MCC in the deeper part (left-hand side; ×40). A higher magnification of the MCC component disclosed solid nests of monotonous neoplastic cells with a fine chromatin pattern and frequent mitoses (**E**), as well as immunoreactivity for synaptophysin (**F**). ((**E**,**F**), ×400). Immunohistochemically, the non-SCC tumor cells also expressed CK20, INSM1, and TTF-1 (clone SP141), confirming the diagnosis of combined MCC and SCC. The tumor cells were negative for Merkel cell polyomavirus (MCPyV) large T antigen (clone CM2B4, Santa Cruz, CA, USA). The patient received reduced-dose combination chemotherapy with cisplatin and etoposide because of his old age. Unfortunately, the patient passed away after undergoing chemotherapy due to a progressive disease within two months.

## 2. Discussion

MCC is a rare and aggressive cutaneous neuroendocrine carcinoma with two major etiologic factors: the integration of the Merkel cell polyomavirus (MCPyV) genome into the host cells and the genetic damage from long-term ultraviolet light exposure [3,10,12,13]. MCPyV infection has been reported to be the etiological agent responsible for up to 80% of MCC in Europe [3]. In our recent study of MCC in Taiwan, 58% (11/19) cases were associated with MCPyV by immunohistochemistry for the large T antigen, and these patients had a better overall survival than those who were MCPyV-negative [14]. Using comprehensive genomic profiling, DeCoste et al. recently showed that MCPyV(+) cases had a lower tumor mutation burden than MCPyV(−) cases, with the latter carrying frequent mutations in *TP53*, *RB1,* and *NOTCH* family genes [10]. On the other hand, ultraviolet-related MCPyV(−) MCC cases include pure or combined MCC, with the latter being reported to be unrelated to MCPyV [7,8,15]. In the study by DeCoste et al., they revealed a substantial mutational overlap between the pure NEC and the SCC components of combined MCC/SCC cases, suggesting that MCPyV(−) MCCs may arise from a keratinocytic precursor lesion or from a tumor stem cell [10]. Despite major advances, the exact pathogenesis of combined MCCs is still not fully elucidated.

MCC is an aggressive skin cancer, with 26% of cases presenting lymph node involvement at diagnosis [3]. We reviewed the literature and identified 13 cases of combined/collision MCC with nodal metastasis as initial presentations (Table 1) [7,8,16,17,18,19,20,21]. As shown in Table 1, SCC was present in most cases of combined/collision MCC, except for one case in which the other tumor component was porocarcinoma. Of note, there were three cases of triple collision tumors, with the non-MCC and SCC component including either sarcoma (n = 2) or basal cell carcinoma (n = 1). We found that metastatic nodal diseases from combined/collision MCC may show divergent differentiation (69%; 9 of 13 cases), and the primary skin tumors were all located in the sun-exposed areas. Interestingly, metastases with a pure MCC component were observed in three combined MCCs in the sun-protected areas (breast, shoulder, and lower back) and one additional case at an unusual site (finger). This observation may reflect the differences in pathogenesis. Furthermore, the median survival time was only 6.5 months among the six patients with available follow-up information. The clinical course was aggressive, consistent with the previous report of a dismal outcome in combined MCCs [7]. Our findings also support the notion that lymphatic draining areas should be scrutinized in MCC patients, particularly in those with combined MCC, for potential nodal metastasis.

MCC is frequently misdiagnosed initially or misinterpreted as other epithelial malignancies, leading to inappropriate or delayed management [3]. Our case illustrates that the diagnosis of combined MCC can sometimes be challenging. A correct diagnosis of a combined tumor might be missed if the initial biopsy sample is too small or superficial and shows only the non-NEC component, as in our case. For the diagnosis of combined MCCs, a larger or excisional specimen or punch biopsy to include the deep dermal component is recommended, as limited biopsy material would lead to a misdiagnosis of a less aggressive skin tumor with a better prognosis [6,9].

## Figures and Tables

**Table 1 diagnostics-13-00449-t001:** The clinicopathological features of combined MCCs with nodal metastases in the literature.

Case	Sex	Age	Location of Primary Skin Tumor	Pathology of Skin Tumor	Site of Metastatic Node	Pathology of Metastatic Node	Follow-Up (m)	Reference
1	94	M	Left upper arm	Triple collision tumor composed of MCC, SCC and sarcoma	Left axillary	MCC and sarcoma	N/A	Hwang, J.H. et al. (2008) [21]
2	88	F	Cheek	MCC and invasive SCC	Submandibular	Metastatic SCC	DOD (6)	Martin, B. et al. (2013) [7]
3	78	M	Preauricular	MCC and invasive SCC	Cervical	MCC and SCC	N/A	Martin, B. et al. (2013) [7]
4	79	M	Right temple	Triple collision tumor composed of MCC, SCC and sarcoma	Right parotid and postauricular	MCC and SCC	DOD (2)	Martin, B. et al. (2013) [7]
5	N/A	N/A	N/A	MCC and porocarcinoma	Unspecified	MCC and porocarcinoma	N/A	Martin, B. et al. (2013) [7]
6	80	M	Right middle finger	MCC and invasive SCC	Right axillary	MCC	DOD (5)	Ansai, S. et al. (2015) [20]
7	77	F	Right breast	MCC and invasive SCC	Right axillary	MCC	N/A	Chou, T. C. et al. (2016) [19]
8	49	M	Right forearm	MCC and invasive SCC	Right axillary	MCC and SCC	DOD (N/A)	Navarrete, J. et al. (2018) [18]
9	51	M	Left ear	MCC and invasive SCC	Left retropharyngeal and deep parotid	MCC and SCC	N/A	Suaiti, L. et al. (2019) [17]
10	66	M	Right anterior shoulder	Triple collision tumor composed of MCC, BCC and SCC in situ	Right axillary	MCC	N/A	Hobbs, M.M. et al. (2020) [16]
11	83	M	Right leg	MCC and invasive SCC	Left inguinal	MCC and SCC	DOD (<12)	Ríos-Viñuela, E. et al. (2022) [8]
12	78	M	Neck	MCC and invasive SCC	Cervical	MCC and SCC	DOD (12)	Ríos-Viñuela, E. et al. (2022) [8]
13	79	M	Lower back	MCC and invasive SCC	Left inguinal	MCC	DOD (2)	Current case

Abbreviation: MCC—Merkel cell carcinoma; SCC—squamous cell carcinoma; BCC—basal cell carcinoma; DOD—died of disease; N/A—data not available; and M—month.

## Data Availability

Data are available on request due to all institutional restrictions related to patient privacy.

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
