# Peer review of "Combined Merkel Cell Carcinoma with Nodal Presentation: Report of a Case Diagnosed with Excisional but Not Incisional Biopsy and Literature Review"

_diagnostics, 2023, doi:10.3390/diagnostics13030449_

Round 1

Reviewer 1 Report

The authors have analyzed  skin cancer. This work has a good potentiality.  The overall studies can draw reader’s attention. I would suggest to add  10.1007/s11071-020-05781-6.  I do recommend for publication.

Author Response

Response to reviewer #1:

The authors have analyzed skin cancer. This work has a good potentiality.  The overall studies can draw reader’s attention. I would suggest to add 10.1007/s11071-020-05781-6.  I do recommend for publication.

Reply: We thank the reviewer for the favorable comments. However, there seems to be a mistake in the information provided for a reference to add. The number https://doi.org/10.1007/s11071-020-05781-6 leads to an related paper, i.e., Das, P., Mukherjee, S., Das, P. et al. Characterizing chaos and multifractality in noise-assisted tumor-immune interplay. Nonlinear Dyn 101, 675–685 (2020).

Reviewer 2 Report

Well-written case report with relevance to clinical practice. I would suggest that the authors explain the potential common pathogenesis behind collision tumors in MCPyV positive and negative MCC.

Thank you very much

Author Response

Response to reviewer #2:

Well-written case report with relevance to clinical practice. I would suggest that the authors explain the potential common pathogenesis behind collision tumors in MCPyV positive and negative MCC.

Reply: We thank the reviewer for the favorable comments. We have made changes to include the pathogenesis in the first paragraph of Discussion section in the revised manuscript.

Reviewer 3 Report

This work which has been submitted in the “Interesting Images” category of Diagnostics MDPI, is an interesting case of Merkel cell carcinoma, a very rare tumor of neuroendocrine origin, which have been diagnosed in a 79-year-old Taiwanese male combined with nodal metastasis at initial presentation. The authors also reported the currently known cases of MCC with nodal metastasis present in the literature. The strength of the work is the extremely rare case which has been presented. Despite the work being too concise (please see below), it is very interesting, but it needs improvements.  The figures are fine, well designed, and informative. I have several comments:
-    The introduction should be improved. Please include a couple of sentences on MCC epidemiology (doi: 10.1007/s13671-014-0068-z), risk factors (doi: 10.1038/nrdp.2017.77) and therapeutic approaches (doi: 10.1007/s13555-019-0288-z).
-    A more detailed description of the presented case, diagnosis,  co-morbidities, therapy, follow-up would improve the quality of the work.
-    I also suggest enlarging the section mentioning the currently known cases of MCC with nodal metastasis present in the literature. The readers would benefit for a brief presentation of the already published works
-    The sentence in line 30-31 is lacking in supporting reference. These references should be included (PMID: 28174236 and PMID: 34943208)
-    Has the tumor tissue been evaluated for Merkel cell polyomavirus DNA by PCR and LT oncogenic protein by IHC?
-    The subhead “introduction” seems to cover the entire text. Wouldn't it be better to remove it? Or change it with another more appropriate subhead title?

Author Response

Response to reviewer #3:

  1. The introduction should be improved. Please include a couple of sentences on MCC epidemiology (doi: 10.1007/s13671-014-0068-z), risk factors (doi: 10.1038/nrdp.2017.77) and therapeutic approaches (doi: 10.1007/s13555-019-0288-z).

Reply: We have made changes to include the epidemiology, risk factors and therapeutic approaches for MCC in the first paragraph of Background section and cited all three publications mentioned.

  1. A more detailed description of the presented case, diagnosis, co-morbidities, therapy, follow-up would improve the quality of the work.

Reply: We have modified the manuscripts with more detailed clinical and pathological findings according to the reviewer’s suggestions.

  1. I also suggest enlarging the section mentioning the currently known cases of MCC with nodal metastasis present in the literature. The readers would benefit for a brief presentation of the already published works

Reply: We have modified the last 2nd paragraph of the revised manuscript according to the reviewer’s suggestions.

  1. The sentence in line 30-31 is lacking in supporting reference. These references should be included (PMID: 28174236 and PMID: 34943208)

Reply: We thank the reviewer for pointing this out. We have added these two papers as suggested.

  1. Has the tumor tissue been evaluated for Merkel cell polyomavirus DNA by PCR and LT oncogenic protein by IHC?

Reply: The immunohistochemical study showed that the tumor cells are negative for Merkel cell polyoma virus large T antigen (clone CM2B4, Santa Cruz, CA., USA).

  1. The subhead “introduction” seems to cover the entire text. Wouldn't it be better to remove it? Or change it with another more appropriate subhead title?

Reply: We thank the reviewer for this suggestion. The subheading “Introduction” has been changed to “Background”. We have added an additional subheading of “Discussion”.

Round 2

Reviewer 3 Report

The ms can be accepted in the present form.